# What are the effects of implementing patient-controlled admissions in inpatient care? A study protocol of a large-scale implementation and naturalistic evaluation for adult and adolescent patients with severe psychiatric conditions throughout Region Stockholm

Maria Smitmanis Lyle,[1,2] Emelie Allenius,[1,2] Sigrid Salomonsson,[1,2] Anna Björkdahl,[1,2] Mattias Strand,[1,2] Lena Flyckt,[1,2] Clara Hellner,[1,2] Tobias Lundgren,[1,2] Nitya Jayaram-Lindström,[1,2] Alexander Rozental  [1,2,3]

**Correspondence to**
Dr Alexander Rozental; alexander.rozental@ki.se

## ABSTRACT

**Introduction** Patient-controlled admissions (PCAs) represent a change in psychiatric inpatient care where patients are allowed to decide for themselves when hospitalisation might be required. Prior research has demonstrated that PCA increase the number of admissions, but decrease days in inpatient care, while both the admissions to and days in involuntary care decrease. However, investigations have been restricted to specific patient groups and have not examined other possible benefits, such as effects on symptoms, quality of life and autonomy.

**Methods and analysis** This study explores the implementation process and effects of PCA in Region Stockholm, who is currently introducing PCA for all patients with severe psychiatric conditions and extensive healthcare utilisation. In total, the study comprises approximately 45 inpatient wards, including child and adolescent psychiatry. In a naturalistic evaluation, patients assigned PCA will be followed up to 36 months, both with regard to hospitalisation rates and self-reported outcomes. In addition, qualitative studies will explore the experiences of patients, caregivers of adolescents and healthcare providers.

**Ethics and dissemination** Approval has been granted by the Swedish Ethical Review Authority (Dnr: 2020-06498). The findings from this study will be disseminated via publications in international peer-reviewed journals, at scientific conferences, as part of two doctoral theses, and through the Swedish Partnership for Mental Health.

**Trial registration number** NCT04862897.

## STRENGTHS AND LIMITATIONS OF THIS STUDY

⇒ This study will examine hospitalisation rates and self-reported outcomes for all patients assigned patient-controlled admissions (PCAs) in Region Stockholm, Sweden.

⇒ Patient with different psychiatric disorders from approximately 45 inpatient wards with will be included and followed up to 36 months.

⇒ For the first time, the implications of PCA for child and adolescent psychiatry will be investigated.

⇒ Qualitative studies will explore the experiences of patients, caregivers of adolescents and healthcare providers.

⇒ Due to ethical and judicial reasons, this study will not allow randomisation of patients or clusters, affecting the possibility of drawing causal inferences of outcomes.

## INTRODUCTION

Patients with severe psychiatric conditions, such as schizophrenia spectrum disorders, borderline personality disorder and eating disorders, often have care needs that warrant frequent admissions to inpatient care.

Historically, it has been up to the healthcare providers to decide who and when to admit, sometimes referred to as the gate-keeper model. In essence, patients may seek care on their own, but the final decision to permit admission has always been made by a physician.[1] However, recent trends towards patient participation and shared decision-making in healthcare has questioned this practice.[2] In particular, the concept of patient-controlled admissions (PCAs) marks a dramatic shift in authority and increased partnership in psychiatric inpatient care, whereby patients are given the possibility of admitting themselves when needed. The basic premise is to identify and grant PCA to those who might

benefit the most from its use as a way of preventing deterioration and novel bouts of illness and support patient empowerment. The procedures differ somewhat between settings and patient groups, but typically involves three to five inpatient days per stay, with or without restrictions regarding the number of admissions allowed per month.[1]

The effects of PCA have been examined in several studies. Early attempts of implementation were done in the USA and Australia, but the first large-scale investigations were made in the Netherlands, Norway, Denmark and Sweden.[3–10] In sum, research indicates an increase in the number of admissions to inpatient care, but a decrease in the number of days in inpatient care. Meanwhile, both the admissions to and days in involuntary care decrease, suggesting that patients admit themselves more often, but require less fewer admissions of shorter duration, in line with the idea of using PCA to promote independence and increase help-seeking behaviours. However, apart from hospitalisation rates, self-rated outcomes have been far less explored. Aspects such as recovery and well-being do not demonstrate any significant changes, but self-report measures have been few and mainly focused on general functioning.[4 8 9] Yet, qualitative studies imply that PCA can be beneficial in terms of increasing autonomy, agency, and well-being among patients. This is evident even in those cases where admissions are seldom or never used, suggesting that it might constitute a safety net for those at risk.[11]

PCA has been available since 2014 in Region Stockholm's psychiatric care, which is Sweden's largest hospital sector, but only for specific patient groups; schizophrenia spectrum disorders,[7] eating disorders,[9] and borderline personality disorder.[12] Different procedures for using PCA have also been used depending on diagnosis, such as the length of each stay. However, following a resolution by the popularly elected Region Stockholm Assembly, the implementation of PCA will be completed in 2023 and involve patients with severe psychiatric conditions, irrespective of diagnosis, using the same procedures in the whole region. This will allow a large-scale implementation and evaluation of its effects, both in terms of hospitalisation rates and self-reported outcomes, up to 36 months following the assignment to PCA. A large number of patients with many different diagnoses will take part in several quantitative studies; including substance use, which has never been the subject to evaluation of PCA before. Furthermore, the use of additional self-report measures will permit an examination of its impact on symptoms, quality of life and autonomy. In addition, the implementation of PCA in child and adolescent psychiatry will be the first of its kind, making it possible to examine its potential effects among adolescents. Lastly, qualitative studies will examine the experiences of patients, caregivers of adolescents, and healthcare providers in an attempt to understand the use, and potential benefits and drawbacks of PCA. Overall, approximately 45 inpatient wards will eventually be included in this study, thus being, to the best of the authors' knowledge, the largest attempt at defining the effects of PCA.

## Aims and objectives

The overall aim of this study is to assess the effects of implementing PCA throughout Region Stockholm, using a naturalistic prospective study design. This involves a number of quantitative studies which intend to understand the impact of PCA on hospitalisation rates and self-reported outcomes when introduced in psychiatric care. Qualitative studies will also be conducted to explore the experiences of both patients and healthcare providers with regard to its effects and use, including the perspective of caregivers of adolescents.

### Primary research question

For patients with severe psychiatric conditions, identified as having extensive care needs and who have been assigned PCA by their healthcare providers; how will their hospitalisation rates change during the following 12, 24 and 36 months with regard to the number of admissions and days in inpatient care, and the number of admissions and days in involuntary care?

### Secondary research question

1. Will hospitalisation rates differ between the various patient groups who have been assigned PCA?
2. Will self-reported outcomes change following the assignment of PCA, that is, symptoms, quality of life and autonomy?
3. Will self-reported outcomes differ between the various patient groups who have been assigned PCA?
4. How are the healthcare expenditures for patients who have been assigned PCA affected, and do they differ between the various patient groups?
5. What are the experiences of patients who have been assigned PCA for at least 12 months?
6. What are the experiences of healthcare providers who have worked with PCA for at least 12 months?
7. For adolescents, what are the experiences of caregivers to patients who have been assigned PCA for at least 12 months?
8. For adolescents, will self-reported outcomes change in terms of their caregivers' perception of the patients' symptoms and quality of life?

In line with prior research, this study hypothesises that (1) the hospitalisation rates of patients will increase in terms of number of admissions to inpatient care, but (2) decrease in terms of days in inpatient care, as well as number of admissions and days in involuntary care. Furthermore, self-reported outcomes are expected to demonstrate (3) a decrease in symptoms and (4) increase in quality of life and autonomy. Lastly, qualitative investigations are believed to give a greater understanding of how patients, caregivers of adolescents and healthcare providers experience the use of PCA.

## METHOD AND ANALYSIS
### Setting

This study is placed within the psychiatric care of Region Stockholm, which is the largest hospital sector in Sweden,

serving a population of two million citizens. It is organised as nine public healthcare clinics, and four private healthcare clinics. One of these clinics constitute child and adolescent psychiatry, two concern eating disorders and two are dedicated to substance use disorders. One additional clinic is reserved for forensic psychiatry, but because its patients have been sentenced to care, these are not eligible to receive PCA. The 13 clinics that are included consist of approximately 45 inpatient wards and about 120 outpatient units that work together as pairs (ie, more than one outpatient unit can share the same inpatient ward). Each inpatient ward will in turn designate at least one bed for PCA. The healthcare providers consist of medical doctors and nurses with or without specialist training in psychiatry, mental health workers, counsellors, psychologists, psychotherapists, social workers, physiotherapists, occupational therapists and students undergoing basic or advanced clinical training in the field of medicine or psychiatry. The starting date of this study was 1 January 2022, and the planned end date is 31 December 2027.

## Patients

In Region Stockholm, PCA is assigned to those patients that have a severe psychiatric condition and who are deemed by their healthcare providers to warrant great care needs. Thus, eligible patients present with a documented history of recurrent admissions to inpatient care, a large number of days in inpatient care, or other indications of future hospitalisation. In order to determine eligibility, the following criteria are used by the healthcare providers:

► Ongoing contact with an outpatient unit.
► An individual care plan and crisis plan.
► At least one episode of inpatient care during the previous 1-year period.
► Expected future need of inpatient care.
► Demonstrating an interest in and understanding the concept of PCA (for adolescents, this also includes their caregivers).

The inpatient wards include patients from general psychiatry (eg, schizophrenia spectrum diagnoses, neuropsychiatric disorders and anxiety disorders), geriatric psychiatry, affective disorders (eg, bipolar disorder), substance use disorders, personality disorders, eating disorders, and child and adolescent psychiatry (ie, up to 18 years of age).

## Procedure

Patients who are eligible to receive PCA are invited to a scheduled meeting with their healthcare providers from the outpatient unit and inpatient ward. The purpose and practice of PCA are then discussed verbally and information is provided in writing. Should the patient accept the offer, an agreement on the use of PCA is reached collaboratively and recorded in their medical records. For adolescents, caregivers also have to agree. Typically, PCA involves up to 4 days of inpatient care at a time, up to three times per month, although exceptions to the rule can sometimes be made. The agreement only covers 12 months at a time, after which it can be renewed at the yearly assessments. After the meeting, the patient receives all of the details in text and a printed copy of the agreement.

Each inpatient ward has allocated at least one inpatient bed for patients assigned PCA. Should a patient with an agreement wish to be admitted, they only have to contact the inpatient ward, any time of the day, any day of the week. In case the inpatient bed is already occupied, the patient is put on a waiting list. If more immediate care is required, the patient can always be admitted as usual. All admissions are managed by a nurse with delegated responsibility from the head of clinic. During admission, the patient is assessed with regard to suicidality, violence, and vital parameters, such as blood pressure and body temperature. Should there be an increased risk (eg, suicidal ideation and physical injury), regular intake is made. Once admitted, patients are offered care according to their specific needs and individual care plan. Consultation with a medical doctor is possible, but not part of usual practice. The patient is also free to discharge at will, which is also handled by a nurse; however, should the patient warrant further stay at the inpatient ward, a medical doctor can decide if regular admittance is necessary.

Since PCA is part of regular healthcare, patients can always agree to its use but refuse to participate in the present research study. The offer to take part in a study on the effects of PCA is always made after the use of PCA first has been agreed on, whereupon the objectives of this study is discussed verbally and information is provided in writing. Informed consent is required and documented in the medical records. For adolescents, informed consent needs to be signed by both the adolescent and all caregivers (eg, legal parents, custodians or other entities responsible for the patient's care). Following this, the patient completes the self-reported outcome questionnaires that are included in the study, as reviewed below. For adolescents, this also involves the caregivers' perception of the patient's well-being. All of the self-reported outcome questionnaires are then completed at the mandatory yearly assessments, except for the Visual Analogue Scales (VAS) that are administered each month via SMS (more information provided under self-reported outcomes). This is in line with the procedures surrounding the use of PCA in Region Stockholm, where mandatory yearly assessments are made to check-up on patients' health status and whether or not PCA should be continued. This is also believed to improve data collection as it does not require paper-and-pencil administration or letters.

The implementation of PCA throughout Region Stockholm is coordinated and supported by the Centre for Psychiatry Research, which is a R&D centre run jointly by Karolinska Institutet and the hospital sector. The Centre for Psychiatry Research performs an assessment of the

risks and needs of each outpatient unit and inpatient ward, and provides training and supervision regarding all of the PCA routines. This consists of an online course and a half-day on-site workshop, and the latter of regular meetings with all of the healthcare providers. The Centre for Psychiatry Research is also responsible for evaluating the effects of PCA and has a research steering group consisting of the authors of the current study protocol. These range from doctoral candidates to senior-level researchers with comprehensive experience of conducting both quantitative and qualitative studies in a psychiatric setting.

## Study design

This study will evaluate the effects of PCA in Region Stockholm in collaboration with the Commissioner of Healthcare. This follows a resolution made by the popularly elected Region Stockholm Assembly, which means that both the implementation and policy surrounding PCA have already been decided. Hence, according to the decision, all patients with severe psychiatric conditions who are considered by their healthcare providers to have great care needs will be offered PCA as long as the criteria mentioned above are fulfilled. Furthermore, because the primary objective of the PCA programme is the provision of healthcare and not research, a more complex study design using randomisation on an individual or clinic level (ie, cluster) is not allowed due to ethical and judicial reasons. Hence, this study is restricted to a naturalistic prospective study design. Patients are followed for 12, 24 and 36 months using their medical records to check changes in hospitalisation prior to (pre) and following inclusion in the PCA programme (post), similar to what has been done previously.[7] For example, patients who are included in November 2021 may have hospitalisation rates retrieved from the period November 2018–2020 and compared against hospitalisation rates from the period November 2022–2024, depending on the availability of historical data. In terms of self-reported outcomes, patients complete the measures when being assigned PCA and at the yearly follow-up assessments, with the exception for the VAS that is administered monthly via SMS. However, in these specific cases, historical data do not exist as these measures have not been collected previously. Only patients with no previous use of PCA when they consent to take part in this study will be analysed, as recorded in their hospital records.

## Patient and public involvement

This study is conducted in collaboration with the Swedish Partnership for Mental Health (NSPH). Patient and public representatives are not involved in issues surrounding study design, but are informed about and actively engaged in formulating research questions, determining outcomes and disseminating the results.

## Demographic variables

Demographic variables are recorded in the patients' medical records through their social security number. These are age, gender, civil status and children. Psychiatric disorders are entered manually by healthcare providers using the Swedish version of the International Classification of Diseases and Related Health Problems, 10th Revision.

## Hospitalisation rates

Hospitalisation rates are recorded in the patients' medical records when admitted to inpatient care or involuntary care (ie, when and how long each stay lasted). This consists of number of admissions and days in inpatient care, and number of admissions and days in involuntary care. All of the healthcare providers have been trained in how to enter this information in the system, including coding schemes specifically designed for this study in order to identify and track patients.

## Self-reported outcomes

Self-reported outcomes (ie, questionnaires) are collected after the patients have been assigned PCA and consented to take part in the research. For an overview, see table 1.

EQ5D-5L: A five-item questionnaire on current health status on the dimensions; mobility, self-care, usual activities (eg, family), pain/discomfort and anxiety/depression. Each item is scored in relation to a five-step scale, for example, 'I am not anxious or depressed' (1) to 'I am extremely anxious or depressed' (5). Furthermore, EQ5D-5L also includes a VAS on current health status; 'worst imaginable health' (0) to 'Best imaginable health' (100).[13]

WHO Disability Assessment Schedule 2.0: A 12-item questionnaire on overall physical and psychological well-being, for example, 'In the past 30 days, how much difficulty did you have in: Taking care of household responsibilities?'. It is scored on five-point Likert-scale, 'None' (1) to 'Extreme or cannot do' (5). Three additional items also concern the number of days difficulties were present or interfered with daily living.[14]

Clinical Global Impression (CGI): A clinician-rating scale that asks healthcare providers to rate the health status of a patient at an initial assessment (referred to CGI-S); 'normal, not at all ill' (1) to 'among the most extremely ill patients' (7). The CGI is also used to determine the improvement of a patient (referred to as CGI-I); 'very much improved since the initiation of treatment' (1) to 'very much worse since the initiation of treatment' (7).[15]

Brunnsviken Brief Quality of Life Scale (BBQ): A 12-item questionnaire on the quality of life in six domains: leisure time, view of life, creativity, learning, friends and friendship, and myself as a person. Each domain is scored in accordance with satisfaction and importance, 'do not agree at all' (0) and 'agree completely' (4), which are then multiplied. The sum of the six products constitutes the total quality of life score.[16]

General Self-Efficacy Scale: A 10-item questionnaire on self-efficacy, that is, the belief to succeed in a given situation or task, for example, 'I can always manage to

**Table 1** Overview of the self-reported outcomes

| Self-report measure | Assessment | | | | |
| --- | --- | --- | --- | --- | --- |
| | Assigned PCA | Monthly measures | 12 months | 24 months | 36 months |
| Adults | | | | | |
| EQ5D-5L | X | | X | X | X |
| WHODAS | X | | X | X | X |
| CGI* | X | | X | X | X |
| BBQ | X | | X | X | X |
| GSE | X | | X | X | X |
| GAD-7 | X | | X | X | X |
| PHQ-9 | X | | X | X | X |
| VAS | X | X | X | X | X |
| Adolescents | | | | | |
| KIDSCREEN | X | | X | X | X |
| ASSIST-Y | X | | X | X | X |
| C-GAS* | X | | X | X | X |
| SDQ† | X | | X | X | X |
| RCADS† | X | | X | X | X |

*Clinician-rating scale.
†Also completed by caregivers.
ASSIST-Y, Alcohol, Smoking and Substance Involvement Screening Test; BBQ, Brunnsviken Brief Quality of Life Scale; C-GAS, Children's Global Assessment Scale; CGI, Clinical Global Impression; GAD-7, Generalised Anxiety Disorder-7 Items; GSE, General Self-Efficacy Scale; PCA, patient-controlled admissions; PHQ-9, Patient Health Questionnaire-9 Items; RCADS, Revised Children's Anxiety and Depression Scale; SDQ, The Strengths and Difficulties Questionnaire; VAS, Visual Analogue Scales; WHODAS, WHO Disability Assessment Schedule 2.0.

solve difficult problems if I try hard enough' (Item 1). It is scored on a four-point Likert scale, 'not at all true' (1) to 'exactly true' (4).[17]

Generalised Anxiety Disorder-7 Items: A seven-item questionnaire on symptoms of anxiety and worry, for example, 'How often have they been bothered by the following over the past 2 weeks? Feeling nervous, anxious, or on the edge' (Item 1). It is scored on a four-point Likert scale, 'not at all' (0) to 'nearly every day' (3).[18]

Patient Health Questionnaire-9 Items: A nine-item questionnaire on symptoms of depression and mood, for example, 'How often have they been bothered by the following over the past 2 weeks? Little interest or pleasure in doing things?' (Item 1). It is scored on a four-point Likert scale, 'not at all' (0) to 'nearly every day' (3).[19]

Visual Analogue Scales (VAS): Four items assessing different aspects of the patients' care needs and current health status are administered monthly via SMS. These include the following: (1) 'I feel confident about receiving care when I need it', (2) 'I am able to actively participate in my care' and (3) 'I believe care is available when needed'. These are scored on a continuum ranging from 'I do not agree at all' (0) to 'totally agree' (10). Also, a fourth item concerns 'How would you rate your current health status?', which is rated between 'worst possible health' (0) and 'best possible health' (10).

For adolescents and their caregivers, a separate set of self-reported outcomes will be used.

KIDSCREEN: An 11-item questionnaire on aspects related to health-related quality of life, for example, 'Thinking about the last week: have you been able to do the things that you want to do in your free time?' (Item 1). It is scored on a five-point Likert scale, 'not at all' or 'never' (0) to 'extreme' or 'always' (4). One item is supposed to reflect the participant's overall health status, that is, 'In general, how would you say your health is?'. KIDSCREEN is administered to both the patients and their caregivers.[20]

Alcohol, Smoking and Substance Involvement Screening Test: A seven-item questionnaire on alcohol, substance, and tobacco use among adolescents. It screens for the consumption and negative effects of different compounds, similar to AUDIT and DUDIT, weighting scores differently depending on frequency and impact of their use.[21]

Children's Global Assessment Scale: A clinician-rating scale that asks healthcare providers to rate the health status of a patient. It is scored on a continuum ranging from 'Needs constant supervision' (1-10) to 'Superior functioning' (100).[22]

The Strengths and Difficulties Questionnaire (SDQ): A 25-item questionnaire on the psychological well-being of an adolescent, for example, 'Often loses temper' (Item 5). It is scored on a three-point Likert scale, 'not true' (0) to 'certainly true' (2). SDQ is administered to both the patients and their caregivers.[23]

Revised Children's Anxiety and Depression Scale (RCADS): A 47-item questionnaire covering different symptoms of different psychiatric conditions, for example, 'I worry that something bad will happen to me' (Item 27). It is scored on a four-point Likert scale, 'never' (0) to 'always' (3). RCADS is administered to both the patients and their caregivers.[24]

A separate study will also investigate how the use of PCA among adolescents affect their caregivers' quality of life (BBQ), functioning (Burden Assessment Scale),[25] and the perceived reception by healthcare providers (Family Involvement and Alienation Questionnaire).[26]

## Sample size calculation

This study intends to examine the impact of introducing PCA in psychiatric care in Region Stockholm. As such, only patients who are assigned PCA and have consented to take part in the research are included. An a priori sample size calculation, which is typically used to estimate an adequate sample size for subsequent recruitment, is thus not informative. However, based on previous experiences of introducing PCA in the region, it is reasonable to expect about twelve patients per inpatient ward who consent to participate, which would amount to a total sample size of 564.[7] When calculating the standardised effect size possible to detect under these circumstances, based on the difference between two dependent means, and using a probability level of 0.05 and a power of 0.80, the result is Cohen's $d$ of 0.12, which is a small, although clinically relevant effect for a population of patients with severe psychiatric disorders, and in line with similar findings on self-reported outcomes.[4 8 9]

## Statistical analysis

Previous studies on the use of PCA have demonstrated that hospitalisation rates are seldom normally distributed. In essence, few patients will use the possibility to admit themselves to inpatient care, resulting in data that is positively skewed and has a leptokurtic distribution. Investigating whether assumptions of normality are violated will therefore be essential prior to any statistical analysis, whereby suitable means of analysing the data will be employed, that is, non-parametric tests or bootstrapping procedures. As for the self-reported outcomes, no prior evidence seems to suggest that data will not be normally distributed. Here, conditional changes over time will instead be modelled using a mixed models-approach, that is, multilevel models. All analyses will be made according to an intention-to-treat principle, accounting for those values that have been lost via multiple imputation or maximum likelihood estimation. Likewise, all analyses will control for potential confounders, such as hospitalisation rates prior to PCA and self-reported outcomes when assigned PCA, as well as demographics available in the hospital records, for example, age, gender, civil status, children and psychiatric disorders.

Apart from studying the results of introducing PCA, this study will also assess its cost-effectiveness, similar to a newly published study from Denmark.[10] This includes determining the direct and total healthcare costs prior to and after being assigned PCA. Direct costs are the expenditures associated with each visit to a psychiatric emergency department, inpatient care and involuntary care. Such expenditures are calculated from standardised price estimates provided by Region Stockholm, resulting in an approximation of how expenditures change following the implementation of PCA. Total healthcare costs are the expenditures of other types of healthcare, for example, visits to primary care. Moreover, cost–utility analyses will also entail so-called quality-adjusted life-years (QALYs), which represent the improvement in terms of well-being for each incremental increase in costs by allowing patients to admit themselves, that is, a cost–utility approach. QALYs can be derived from self-reported outcomes on the EQ5D-5L, which is commonly used for this purpose, using population-generated preference weights which are available for a Swedish setting.[27] This will be compared to matched patients using registries. In addition, probability sensitivity analyses will be conducted to determine the errors that surrounds the cost-utility calculations made. This provides confidence ellipses at 50%, 75% and 95% that represent the uncertainty associated with each estimate.

## Qualitative studies

In addition to the quantitative studies, this study will also involve a number of qualitative investigations to further the current understanding of its use and effects, similar to previous research in the field.[28 29] First, patients who have been assigned PCA for at least 12 months will be recruited to share their perspective (n=30). Recruitment will be purposeful to ensure heterogeneity with regard to patient groups and the extent to which PCA have been used. The aim is to explore how PCA has been perceived and what impact it might have had on both day-to-day activities and quality of life. Second, healthcare providers from different inpatient wards and professional backgrounds, having had at least 12 months of experience of working with PCA, will be recruited to provide their view of its application (n=30). The aim is to investigate their attitudes towards the method in general, the implementation of PCA in particular, and whether these have changed over time as a result of working with PCA. Third, caregivers of adolescents who have been assigned PCA for at least 12 months will be recruited to share their experiences (n=20), both in terms of how they have perceived its effects for the adolescent and how it might have affected their own situation. All of the qualitative studies will be made using individual semi-structured interviews that are recorded digitally and transcribed ad verbum. Given the exploratory nature of this research, all analyses will be inductive and done in accordance with thematic analysis.[30] This approach is often used in social sciences as well as in health sciences and helps to understand the viewpoint of an individual in relation to a specific topic, phenomenon or subject matter. This can be particularly

useful in examining patterns and concepts in shared responses, which is helpful in generating hypotheses in a research field that lacks prior research.

## ETHICS AND DISSEMINATION

This study was approved in March 2021 by the Swedish Ethical Review Authority (Dnr: 2020-06498), and has been registered as a clinical trial on www.ClinicalTrials.gov (NCT04862897). All data are recorded using Region Stockholm's public healthcare's digital medical records system, TakeCare, and only exported for statistical analyses in an aggregated pseudonymised format. Moreover, participant identifiers instead of Swedish social security numbers are used to ensure anonymity, for example, LSR64UFL. To participate in the research, informed consent has to be provided, and for adolescents, this includes caregivers (eg, legal parents, custodians or other entities responsible for the patient's care).

Given the novelty of the intervention and lack of research concerning PCA, this study has the potential to make a significant contribution to the field. Apart from examining its effects with regard to hospitalisation rates, self-reported outcomes will help to investigate possible benefits on symptoms, quality of life and autonomy. In addition, qualitative studies will provide useful insights regarding its use and implications for care and everyday life, including the opinions of caregivers of adolescents. In order to successfully disseminate the results, findings will be presented in international peer-reviewed journals, at scientific conferences and as part of two doctoral theses. In addition, an ongoing collaboration with patient representatives will ensure patient and public involvement and dissemination to patients and their significant others, primarily through the NSPH.

**Author affiliations**
[1]Centre for Psychiatry Research, Department of Clinical Neuroscience, Karolinska Institutet, Stockholm, Sweden
[2]Stockholm County Council, Stockholm Health Care Services, Stockholm, Sweden
[3]Department of Psychology, Uppsala University, Uppsala, Sweden

**Contributors** AR, SS, TL, NJ-L, MSL, EA, AB, CH, MS and LF have all been involved in the design and planning of this study. All authors have taken part in drafting the manuscript and are also part of the steering group for researching PCA in the region.

**Funding** The present study, its implementation, and research is funded by the Commissioner of Healthcare in Region Stockholm and Vetenskapsrådet (2021-06378).

**Competing interests** None declared.

**Patient and public involvement** Patients and/or the public were not involved in the design, or conduct, or reporting, or dissemination plans of this research.

**Patient consent for publication** Consent obtained directly from patient(s)

**Provenance and peer review** Not commissioned; externally peer reviewed.

**ORCID iD**
Alexander Rozental http://orcid.org/0000-0002-1019-0245

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
