## [Reviewer comments · BMJ Open]

ARTICLE DETAILS

TITLE (PROVISIONAL)	What are the effects of implementing patient-controlled admissions in inpatient care? A study protocol of a large-scale implementation and naturalistic evaluation for adult and adolescent patients with severe psychiatric conditions throughout Region Stockholm
AUTHORS	Smitmanis Lyle, Maria; Allenius, Emelie; Salomonsson, Sigrid; Björkdahl, Anna; Strand, Mattias; Flyckt, Lena; Hellner, Clara; Lundgren, Tobias; Jayaram-Lindström, Nitya; Rozental, Alexander

VERSION 1 – REVIEW

REVIEWER	Lakra, Vinay Melbourne Health, North West Area Mental Health Services
REVIEW RETURNED	19-Nov-2021

GENERAL COMMENTS	Very well designed study to answer a complex question. The study includes qualitative interviews with carers of adolescents. It would have been better informed to also include another arm - carers / family members of adult patients, as some of them would be involved in supporting or helping their family member despite the patient being an adult. Their views are also important to provide a comprehensive picture.
--

REVIEWER	Paaske, Louise University of Southern Denmark, Department of Public Health
REVIEW RETURNED	29-Nov-2021

GENERAL COMMENTS	Thank you for the opportunity to review this protocol. Overall, it is well written, and sets out to answer important questions in the field of research concerning patient-controlled admissions among patients with severe psychiatric conditions. Especially to estimate how PSI affects Quality Adjusted Life Years (QALYs), which has not been assessed yet. However, the study design can be reconsidered with advantage taking a controlled study design into account. Considerations about lacking the use of a controlled study design should at least be mentioned. Further there is no specifications on how to adjust the analysis for potential confounding. With a revision of the study design and the statistical analysis plan it could make an important and valuable contribution to the field. Major comments - Study design The study design is presented as a naturalistic prospective study design comparing pre- and post-hospitalization rates, whereas self-reported outcomes will be measured at assignment and at the
--

	yearly follow-up assessments with the exception for VAS to be administered monthly. Can you please elaborate on the rationale for using this design? How will you ensure that no 'contaminated' patients will be included in the sample? From how I read the protocol, the PCA has been slowly implemented across some of the hospitals, but now will be rolled out everywhere in Stockholm. How will you ensure that the patients you study are not already using it? Why not consider a stronger quasi-experimental design including a control group, when the data collection will proceed prospectively? For example, a cluster randomization comparing with a comparable group of psychiatric patients at another hospital in Sweden were PCA is not introduced? Both the effect and costs can be driven by different factors than PCA which is the biggest problem with the present study design. Using a controlled study design will enable adjustment for different potential confounding and trends over time. If you are sticking with the design, at least justify why you do not include a control group. - Statistical analysis The section of statistical analysis does not present how to handle potential confounding. Such as: demographics, duration of disease, disease development, socio-economic factors, or other treatment beside PSI. Given the proposed study population, it is reasonable to imagine there will be challenges with completeness of PRO data. Please provide a plan for how missing data will be handled. - Cost-effectiveness Direct costs (including the expenditures associated with each visit to a psychiatric emergency department, inpatient care, and involuntary care) is a very narrow perspective. It is possible that patient's consumption of medicine and/or health services outside of psychiatry changes they are assigned to PCA. Therefore you might consider including the total health care costs, instead of only the direct psychiatric costs. If you do not expand the perspective, at least explain/justify why total health care costs will not be included. Using a before and after design only including PCA patients as their own control is not a feasible approach to assess a cost-effectiveness and a cost-utility (estimating QALY's) analysis. Providing a control group will raise the internal validity considerably and make all the analysis reach a higher scientific level. Minor comments - Introduction; page 6, line 22-37 For your help there has recently been published a relevant Danish study assessing patient-controlled admissions effects on the total healthcare cost among psychiatric patients with severe mental disorders (Paaske et al., 2021). Regarding the qualitative part of the protocol Ellegaard et al. have contributed a lot to this field (Ellegaard, Mehlsen, et al., 2017) (Ellegaard et al., 2020) (Ellegaard et al., 2018) (Ellegaard, Bliksted, et al., 2017). - Self-reported outcomes; page 10, line 20 ->
--	--

	Self-reported outcomes will be collected through 8 different questionnaires among adult patients and through 5 different forms among the adolescent. The population consists of patients with severe psychiatric conditions. Please explain how you will ensure complete data collection on PROs with such a population. You simply need to elaborate on the specific procedure of how you will collect the PROs (online, paper, mail?) and how you will contact/remind, how often reminders will be sent, etc. - Statistical analysis; page 13, line 10 If you are using QALYs, it is cost-utility, not cost-effectiveness analysis References Ellegaard, T., Bliksted, V., Lomborg, K., & Mehlsen, M. (2017). Use of patient-controlled psychiatric hospital admissions: patients' perspective. Nord J Psychiatry, 71(5), 370-377. https://doi.org/10.1080/08039488.2017.1302505 Ellegaard, T., Bliksted, V., Mehlsen, M., & Lomborg, K. (2018). Integrating a Patient-Controlled Admission Program Into Mental Health Hospital Service: A Multicenter Grounded Theory Study. Qual Health Res, 28(6), 888-899. https://doi.org/10.1177/1049732318756301 Ellegaard, T., Bliksted, V., Mehlsen, M., & Lomborg, K. (2020). Feeling safe with patient-controlled admissions: A grounded theory study of the mental health patients' experiences. J Clin Nurs, 29(13-14), 2397-2409. https://doi.org/10.1111/jocn.15252 Ellegaard, T., Mehlsen, M., Lomborg, K., & Bliksted, V. (2017). Use of patient-controlled psychiatric hospital admissions: mental health professionals' perspective. Nord J Psychiatry, 71(5), 362-369. https://doi.org/10.1080/08039488.2017.1301548 Paaske, L. S., Sopina, L., Olsen, K. R., Thomsen, C. T., Benros, M. E., Nordentoft, M., & Hastrup, L. H. (2021). The impact of patient-controlled hospital admissions among patients with severe mental disorders on health care cost: A nationwide register-based cohort study using quasi-experimental design. J Psychiatr Res, 144, 331-337. https://doi.org/10.1016/j.jpsychires.2021.10.032 Acknowledgement I thank Liza Sopina for helping me completing the review.
--	---

VERSION 1 – AUTHOR RESPONSE

Reviewer 1:

1. Very well designed study to answer a complex question. The study includes qualitative interviews with carers of adolescents. It would have been better informed to also include another arm - carers / family members of adult patients, as some of them would be involved in supporting or helping their family member despite the patient being an adult. Their views are also important to provide a comprehensive picture.

We thank the reviewer for the support of our research. The idea to also investigate caregivers or family members (as well as healthcare providers) is important and would add another perspective to the investigation of outcomes of Patient-Controlled Admissions (PCA). This is already included as part

of the qualitative studies that are planned, which will be used to triangulate the results obtained from patients. Moreover, for adolescent patients, a separate study exploring the self-reported changes in health and quality of life for their caregivers will be done, but this is not part of the pre-registered trial that is reported in the study protocol. However, we have added a sentence about this in the manuscript.

Reviewer 1:

1. The study design is presented as a naturalistic prospective study design comparing pre- and post-hospitalization rates, whereas self-reported outcomes will be measured at assignment and at the yearly follow-up assessments with the exception for VAS to be administered monthly. Can you please elaborate on the rationale for using this design?

We thank the reviewer for this comment. As pointed out in the manuscript, a decision to implement PCA throughout Region Stockholm has already been made, prioritizing rollout rather than research. Studying the possible benefits of PCA is thus of secondary nature, restricting the study design to a naturalistic one. There are also ethical and judicial aspects surrounding this, such as that at patient cannot be withheld PCA given that its part of the regular healthcare, preventing the use of randomization. We realize this could have been clearer to begin with, which is why we have added two sentences about it and the administration procedures surround the self-report measures.

2. How will you ensure that no 'contaminated' patients will be included in the sample? From how I read the protocol, the PCA has been slowly implemented across some of the hospitals, but now will be rolled out everywhere in Stockholm. How will you ensure that the patients you study are not already using it?

Once deemed eligible to receive PCA, the patient is also asked to take part in the study and to provide written informed consent. This is then recorded in the medical records, making it possible to track the patient and export data for analysis. Patients who have used PCA previously will not be included in the evaluation and are possible to locate and flag in the same medical records. We have added a sentence about this for clarification.

3. Why not consider a stronger quasi-experimental design including a control group, when the data collection will proceed prospectively? For example, a cluster randomization comparing with a comparable group of psychiatric patients at another hospital in Sweden were PCA is not introduced? Both the effect and costs can be driven by different factors than PCA which is the biggest problem with the present study design. Using a controlled study design will enable adjustment for different potential confounding and trends over time. If you are sticking with the design, at least justify why you do not include a control group.

As described in the manuscript, and now clarified following comment 1. by the reviewer, randomization is not feasible because of ethical and judicial reasons. This was considered and

discussed with the Commissioner of Healthcare in Region Stockholm, but dismissed. Hence, only a naturalistic prospective study design is allowed. We totally agree with the reviewer that a different setup would have been more suitable, but we still believe it is important to follow through with the study as planned.

4. The section of statistical analysis does not present how to handle potential confounding. Such as: demographics, duration of disease, disease development, socio-economic factors, or other treatment beside PSI.

We thank the reviewer for pointing this out. We have now added a sentence for clarification.

5. Given the proposed study population, it is reasonable to imagine there will be challenges with completeness of PRO data. Please provide a plan for how missing data will be handled.

This is already included in the manuscript, i.e., according to intention-to-treat principle, employing either multiple imputation or maximum likelihood estimation.

6. Direct costs (including the expenditures associated with each visit to a psychiatric emergency department, inpatient care, and involuntary care) is a very narrow perspective. It is possible that patient's consumption of medicine and/or health services outside of psychiatry changes they are assigned to PCA. Therefor you might consider including the total health care costs, instead of only the direct psychiatric costs. If you do not expand the perspective, at least explain/justify why total health care costs will not be included.

We thank the reviewer for pointing this out. We have added a sentence about total healthcare costs as well. This should have been included in the previous version, but was missed.

7. Using a before and after design only including PCA patients as their own control is not a feasible approach to assess a cost-effectiveness and a cost-utility (estimating QALY's) analysis. Providing a control group will raise the internal validity considerably and make all the analysis reach a higher scientific level.

We agree, this was not clear enough in the manuscript. We have added a sentence about using registries for this purpose.

8. - Introduction; page 6, line 22-37. For your help there has recently been published a relevant Danish study assessing patient-controlled admissions effects on the total healthcare cost among psychiatric patients with severe mental disorders (Paaske et al., 2021).

We thank the reviewer for this recommendation. This was not available when the manuscript was submitted, but we have not included it.

9. Regarding the qualitative part of the protocol Ellegaard et al. have contributed a lot to this field (Ellegaard, Mehlsen, et al., 2017) (Ellegaard et al., 2020) (Ellegaard et al., 2018) (Ellegaard, Bliksted, et al., 2017).

We thank the reviewer for pointing this out. We have added a sentence and some of the references to our manuscript to highlight this important line of research.

10. Self-reported outcomes will be collected through 8 different questionnaires among adult patients and through 5 different forms among the adolescent. The population consists of patients with severe psychiatric conditions. Please explain how you will ensure complete data collection on PROs with such a population. You simply need to elaborate on the specific procedure of how you will collect the PROs (online, paper, mail?) and how you will contact/remind, how often reminders will be sent, etc.

Data collection is part of the procedures surrounding the use of PCA, i.e., once it is assigned and at mandatory yearly assessments. This will improve response rates as it is part of their regular healthcare and required for continued use of PCA. This was however not entirely clear in the previous version, which is why a paragraph has been added to the manuscript.

11. If you are using QALYs, it is cost-utility, not cost-effectiveness analysis.

We thank the reviewer for pointing this out, it has now been changed.